# Do both timing and duration of screen use affect sleep patterns in adolescents?

**Sarah Hartley** [1,2]*, **Sylvie Royant-Parola**[1], **Ayla Zayoud**[3], **Isabelle Gremy**[3], **Bobette Matulonga**[3]

1 Réseau Morphée, Garches, France, 2 APHP Hôpital Raymond Poincaré, Sleep Center, Université de Versailles Saint-Quentin en Yvelines, Garches, France, 3 Institut Paris Région, Observatoire Régional de Santé, Paris, France

* sarah.hartley@reseau-morphee.fr

**Data Availability Statement:** All relevant data are within the paper and its Supporting Information files.

**Funding:** The authors received no specific funding for this work.

## Abstract

### Background

Sleep duration has declined in adolescents over the last 30 years and screen use has been identified as a risk factor. Studies have examined the duration of screen use and screen-based activities but have not differentiated between evening and night-time use.

### Methods

Cross sectional questionnaire survey of adolescents recruited in schools. Sleep habits on school nights and weekends, symptoms of insomnia and daytime repercussions were recorded using an online questionnaire administered in the classroom setting. Sleep deprivation (<7 hours in bed /night), school night sleep restriction (≥2 hours difference in sleep duration on school nights vs weekends), excessive sleepiness (score >6 on a visual analogue scale), duration of screen use and timing of screen use (evening vs after bedtime) were determined.

### Results

2513 students (53.4% female, median age 15 years) were included. 20% were sleep deprived and 41% sleep restricted. A clear dose effect relationship in a model controlling for age, sex, school level and sociodemographic class was seen with all levels of night-time screen use on sleep deprivation and sleep restriction (>2 hours use sleep deprivation OR 5.23[3.03–9.00]. sleep restriction OR 2.05[1.23–3.42]) and > 2 hours evening use (>2 hours use sleep deprivation OR 2.72[2.15–3.44] sleep restriction OR 1.69[1.36–2.11]) but not moderate evening use. All night-time use and > 2 hours evening use increased the risk of insomnia, non refreshing sleep, and affected daytime function (daytime sleepiness, lack of energy and irritability).

### Conclusions

Both duration of screen use and timing are associated with adverse effects on sleep and daytime functioning in adolescents. More than 2 hours evening use and all night-time use should be avoided.

**Competing interests:** The authors have declared that no competing interests exist.

## Introduction

Sleep is a vital function that evolves throughout the human lifespan. Restorative sleep is essential for health, cognitive function, and mental health [1]. Sleep is regulated by homeostatic sleep pressure and the circadian drive for sleep [2]. While the former depends on the time spent awake (and is thus sensitive to sleep deprivation) the latter depends on external synchronisers or zeitgebers [3]. These act on the central body clock in the suprachiasmatic nucleus which controls the phase of peripheral clocks in individual cells via melatonin secretion from the pineal gland during the night.

Both sleep duration and the phase of the circadian system are important for sleep timing, sleep quality and daytime vigilance [4]. Sleep need in adolescents is often underestimated: studies have shown that adolescents need on average 9 hours a night, but real life sleep times are often much lower [5] and studies suggest that they been reducing over time [6] although this is debated [7]. Many causes have been advanced for this reduction in sleep time including social pressures linked to school work or friendship groups, the use of stimulants such as caffeine, and finally inappropriate stimulation of the central body clock due to evening light.

Light is the most powerful synchroniser of the central body clock and stimulates the melanopsin receptors of intrinsically photosensitive retinal ganglion cells (ipRGCs) in the retina which convey the signal to the suprachiasmatic nucleus. Melanopsin receptors are exquisitely sensitive to blue light in the 446–477 nm range [8]. Modern screens in smartphones, laptops and tablets have screens emit large quantities of blue light, and this has been shown to modulate the expression of circadian genes [9] and block the secretion of melatonin [10]. This translates to sleep disturbance: reading using a tablet clearly delays sleep onset compared to a book [11], and this effect is blocked by wearing blue light blocking glasses [12]. Children and adolescents are particularly sensitive to the effects of blue light [13], especially if they are little exposed to morning bright light. An additional mechanism is likely to be the mental stimulation provided by screen use: playing a video game or using social networks seem to contribute to poor sleep [14].

The American Academy of Pediatrics, in 2016, recommended limiting screen activities to 2 hours a day, but this limit has been shown to be regularly exceeded [15]. Excessive screen use is defined as >3 hours a day of screen use for social activities: ie those not related to school work, but timing of screen use has not been defined. Studies have shown that excessive screen use is linked to insufficient sleep in adolescents [16] although quantifying use is problematic. It is possible that other consequences, such as obesity [17] and adverse mental health outcomes [18] are mediated by reduced sleep time. Reduced sleep duration and poor quality sleep in adolescents have been shown to lead to inattention in class [19] and poor academic performance [20–22].

We hypothesized that in adolescents both the duration of screen use and the timing of screen use (evening and night-time) would be associated with a reduction in both sleep time and sleep quality with effects on daytime function (daytime sleepiness, lack of energy and irritability). Our aim was to examine the associations between the duration and timing of screen use, sleep and daytime function in healthy adolescents in the community.

## Methods

The Morphee health network (Réseau Morphée) is funded by the regional health authority in the Paris region. It aims to provide information about sleep disorders and improve the management of patients suffering from sleep disorders. We performed a transverse survey based study in adolescents aged from 12–19. Participants were recruited within schools during educational sessions on sleep between 2015 and 2019. Prior to educational sessions led by trained

class teachers, consent was sought from parents. Once consent was obtained, students completed an online questionnaire in class which formed one of the tools for the education sessions. Inclusion criteria were belonging to a school participating in the programme and having signed consent from a parent. No identifying data was available on the questionnaire apart from sex, age to the nearest year, school class and school.

The study was approved by the scientific committee of the Réseau Morphée and by the educational authority in each school (Comité pédagogique et du Comité d'Education à la Santé et la Citoyenneté, CESC). The study was approved by the Commission Nationale Informatique et Liberté (CNIL), 8013081 19/12/2016.

## Sleep related measures

The questionnaire had 50 questions including age (in years), sex, school year, the presence of a screen in the bedroom, timing and duration of screen exposure, screen based activities, bedtime and getting up time during the week and at the weekend, sleep quality and symptoms associated with sleep disorders, anxiety and depression.

Sleep time was calculated from mean bedtime and getting up times both during the week and at weekends, and represents the total available window for sleep, not necessarily the actual sleep time which will be reduced by the time taken to fall asleep and wake periods during the night. Adolescents with a sleep time <7 hours were considered to be sleep deprived, adolescents with a difference of ≥2 hours between their sleep time on schooldays and at the weekend were considered to be sleep restricted during the week. We compared sleep deprivation (sleep time of <7 hours) with sleep restriction (a difference between sleep time at the weekend and during the week of >2 hours) to attempt to compensate for adolescents whose sleep needs were less than 9 hours following the definition of sleep restriction in the study by Lo et al [23].

Information on sleep quality was collected looking for difficulties in falling asleep (with a time >60 minutes considered as abnormal [24]) and the presence of refreshing sleep: on a scale of 1–10, sleep was considered non refreshing if the score was ≤6. Insomnia was defined as difficulties falling asleep or maintaining sleep accompanied by daytime consequences (lack of energy, irritability or daytime sleepiness).

## Daytime functioning

Daytime sleepiness was defined as a score >6 on a visual analogue scale of 1–10 concerning the likelihood of falling asleep in class with a maximum score of 10 (extremely sleepy in class). Students were asked about lack of energy and irritability with a yes/no response.

## Screen exposure

Participants were prompted to supply data on the presence of a screen in the bedroom including computer, television and smartphone. Data on the duration of screen use (computers, smartphones, and games consoles) was collated to provide the total duration of screen exposure. Simultaneous use of several screens was not recorded. Mean duration of screen exposure was estimated by students both after dinner before bedtime and during the night (in bed, after lights out) with 5 possible responses (0 minutes, <30 minutes, <1 hour, 1–2 hours, <2 hours). The type of activity in the evening and during the night was recorded (sending texts, social networks, films/series and video games)

## Statistical analysis

Quantitative variables (sleep duration, bedtime and getting up time) were described by median and interquartile range (IQR) and were compared using the Kruskal-Wallis test. Qualitative

variables such as sex, evening screen use, night-time screen use and symptoms of sleep pathology were described by percentage and analysed by Chi$^2$ tests. For multivariant analysis, evening screen use with 5 possible responses from 0 minutes to >120 minutes were collapsed into 3 groups (<60 minutes, 60–120 minutes and >120 minutes). For night-time screen use, the original 5 groups (0 minutes to >120 minutes) were retained as we considered it possible that lower duration screen exposure during the night was physiologically important.

Logistic regression was used to evaluate the associations between sleep duration, sleep disorders evening screen use and night-time screen use. The analysis was adjusted for sociodemographic variables: age, sex and sociodemographic status, evaluated via school location, as previous studies have shown that these are important [25]. Missing variables were excluded from the analysis. Analysis was performed using SAS version 9.4.

# Results

## Population

The online questionnaire could not be uploaded if data was not complete, 6 participants were excluded for incoherent responses. 2513 students were included: 53.4% were girls with a mean age of 14.3 years (median: 15 years, IQR: [12, 16]). All were in secondary education: 45.7% were in middle school (aged 11–15) and 54.3% in high school (aged >15).

## Sleep habits

Mean sleep time over the entire population was 7.8 hours (median 8 hours; IQR: [7, 9]; SD 1:52) during the school week and 9.75 hours (median 10 hours; IQR: [9, 11]; SD 2:17) at the weekend: students aged <12 years slept longer than the 12–15 and >15 years groups (p<0.0001), and middle school students slept longer than high school students (p<0.0001), see Table 1.

Nearly half of all students took >30 minutes to fall asleep and 10% more than 2 hours. Large differences were noted between school days and weekends, both in mean getting up time (07:05 vs 10:38) and in bedtimes (21:52 vs 00:46). 95% got up between 06:00 and 08:00 on school days, whereas 94% got up after 08:00 at the weekends (with 36,3% after 11H00 and 10% after 13:00). A bedtime after midnight was much more common at the weekend (62% vs 14%).

## Sleep and age

Sleep restriction, defined as a reduction of >2 hours in sleep time on schooldays vs weekends increased significantly with age: 30% in <12, 45% in 12–15 and 46% in >15 (p<0.0001), as did sleep deprivation defined as a sleep time <7 hours: 9% <12, 21% 12–15, 31% >15 (p<0.0001).

**Table 1. Age, sleep time, sleep pathology, and daytime sleepiness.**

|  | <12 years n = 847 | 12–15 years n = 842 | >15 years n = 816 | Total n = 2513 | p | P tendance |
|---|---|---|---|---|---|---|
| Median Sleep time on school days; hours: minutes (SD) | 9:00(1:51) | 8:00 (1:42) | 7:00 (1:36) | 8:00 (1:52) | <0.0001 | <0.0001 |
| Median sleep time on weekends; hours: minutes (SD) | 10:05 (2:35) | 10:04 (1:58) | 9: 57(2:11) | 10:00 (2:17) | <0.0001 | <0.0001 |
| Sleep deprivation on schooldays | 9% | 21% | 31% | 20% | < .0001 | < .0001 |
| Sleep restriction | 30% | 45% | 46% | 41% | < .0001 | < .0001 |
| Insomnia | 19% | 17% | 18% | 18% | 0.45 | 0.87 |
| Difficulty falling asleep | 20% | 14% | 16% | 16% | 0.004 | 0.02 |
| Daytime sleepiness | 3% | 2% | 1% | 2% | 0.043 | 0.012 |
| Poor quality sleep | 36% | 48% | 52% | 45% | < .0001 | < .0001 |

Non refreshing sleep also increased with age affected 36% <12, 48% 12–15 and 45% of >15 (p<0.0001). As expected sleep duration also decreased with age whereas no age difference was found for insomnia or for daytime sleepiness.

## Screen use (Table 2)

65% had at least one screen in their bedroom (computer, television, game console or mobile telephone). Before bedtime 11% read or listened to music, 30% did passive screen based activites (ex. watching a film) 32% did active screen based activities (social networks, video games) and 27% did none of the above. 22% used their smartphone and 40% used other screens before going to bed. 27% reported using screens during the night: see Table 2.

82% woke up at least once during the night and in 33% this was due to a message on their smartphone (text or social network). Owning a smartphone was more common in girls (94.4% vs 90.3% p = 0.0002), but excessive screen use (>2 hours) after dinner was more common in boys (26% vs 18% p<0.0001) as was > 2 hours screen use during the night (4.5% vs 2.6% p = 0.0009).

## Screen use and sleep duration

The presence of a screen in the bedroom (computer, television, game console or mobile telephone) was associated with an increase in sleep deprivation (p<0.0001) (Table 2). Multivariate analysis using logistic regression (Table 3), showed that having a screen in the bedroom and owning a smartphone were both associated with an increased risk of sleep deprivation and sleep restriction (p<0.0001) however this was no longer apparent after adjusting for age, sex, school class and sociodemographic class.

Evening screen use was associated with sleep deprivation: bivariate analysis (Table 2) showed that of participants with a sleep time of <7 hours, 51% used screens for >120 minutes in the evening vs 11% of students with no screen use. Multivariate analysis (Table 4) showed a clear dose effect relationship between the duration of evening screen use, sleep deprivation and sleep restriction compared to the group who used screens for <1 hours (all data in bold are significant with p < 0.0001). After adjustment for age, sex, school class and sociodemographic class moderate evening screen use (60–120 minutes) was only significantly associated with sleep restriction, but > 2 hours of evening screen use was significantly associated with both sleep deprivation OR 2.72 [2.15–3.44] and sleep restriction OR 1.69 [1.36–2.11] (Table 4).

Night-time screen use also showed a significant dose effect relationship both before and after adjustment, with > 2 hours of use being strongly associated with both sleep deprivation OR 5.23 [3.03–9.00] and sleep restriction OR 2.05 [1.23–3.42].

**Table 2. The association between screens and sleep time.**

| | | Sleep duration on school nights | | | | |
|---|---|---|---|---|---|---|
| | | **<7 hours n = 505** | **7–9 hours n = 1183** | **≥9 hours n = 825** | **Total** | **p** |
| | Computer in the bedroom | 64% | 58% | 41% | 46% | < .0001 |
| Duration of screen time in the evening | 0 minutes | 11% | 12% | 18% | 14% | < .0001 |
| | <30 minutes | 9% | 17% | 26% | 18% | |
| | 30–60 minutes | 11% | 19% | 17% | 17% | |
| | 60–120 minutes | 18% | 25% | 17% | 21% | |
| | >120 minutes | 51% | 25% | 21% | 30% | |
| | Owns a mobile telephone | 96% | 95% | 87% | 92% | < .0001 |
| | Nighttime users of screen | 10% | 11.8% | 5.4% | 26.7% | < .0001 |
| | Owns a smartphone | 94% | 94% | 88% | 92% | < .0001 |
| | Television in the bedroom | 44% | 35% | 26% | 34% | < .0001 |

**Table 3. The association between the presence of screens in the bedroom or owning a smartphone on sleep and daytime function adjusted for age, sex and sociodemographic class.**

| | Presence of a screen in the bedroom* N = 1632 (64.9%) | | Owning a smartphone** N = 2324 (92.5%) | |
| --- | --- | --- | --- | --- |
| | Crude OR [IC95%] | Adjusted OR [IC95%] | Crude OR [IC95%] | Adjusted OR [IC95%] |
| Sleep deprivation on schooldays | **2.71 [1.75–4.19]** | 1.19 [0.73–1.91] | **1.96 [1.50–2.54]** | 1.19 [0.87–1.62] |
| Sleep restriction | **2.25 [1.42–3.57]** | 1.57 [0.97–2.55] | **1.54 [1.16–2.04]** | 1.19 [0.87–1.62] |
| Insomnia | 0.83 [0.52–1.30] | 0.97 [0.60–1.57] | 0.94 [0.69–1.29] | 0.99 [0.71–1.39] |
| Lack of energy | 1.09 [0.75–1.59] | 0.80 [0.54–1.20] | **1.49 [1.15–1.92]** | 1.21 [0.92–1.60] |
| Irritability | 1.05 [0.66–1.66] | 1.12 [0.69–1.82] | 1.18 [0.86–1.61] | 1.21 [0.86–1.69] |
| Daytime sleepiness | **1.67 [1.12–2.48]** | 1.13 [0.75–1.71] | **1.89 [1.45–2.45]** | **1.47 [1.11–1.96]** |
| Poor quality sleep | **1.64 [1.12–2.42]** | 1.17 [0.78–1.75] | **1.39 [1.09–1.78]** | 1.09 [0.83–1.43] |

Significant data (p < 0.0001) are shown in bold

*Adolescents having at least one screen il their bedroom are compared with those who do not have a screen in their bedroom considered as the reference class (with OR = 1)

**Adolescents who have a smartphone are compared with those who don't have a smartphone considered as the reference class (with OR = 1)

OR ratio are adjusted for age, school department as a proxy of socio economic level and gender.

## Screens, sleep complaints and daytime functioning

Multivariate analysis showed that the presence of a screen in the bedroom was associated with an increase in daytime sleepiness OR1.67 [CI:1.12–2.48] and non refreshing sleep OR1.64 [CI1.12–2.42], but these were no longer significant once corrected for age, sex, school class and sociodemographic class (Table 3). Owning a smartphone was associated with decreased sleep quality OR1.39 [CI:1.09–1.78] and an increase in daytime sleepiness OR1.89 [CI:1.45–2.45] but only daytime sleepiness OR1.47 [1.11–1.96] remained significant once the analysis was adjusted (Table 3). No association was seen with insomnia or irritability.

Evening screen use showed a clear dose effect relationship (Fig 1 and Table 4) persisting after adjustment for age, sex, school class and sociodemographic class with certain elements of daytime function. Both moderate (60–120 minutes) and > 2 hours of evening screen use increased daytime sleepiness OR 2.17 [CI1.75–2.69]. More than 2 hours of evening screen use but not moderate screen use (60–120 minutes) was associated with non refreshing sleep OR 1.60 [1.29–1.98] insomnia 1.60 [1.23–2.00], irritability OR 1.64 [1.28–2.09], and lack of energy OR1.54 [1.24–1.90]. When adolescents who used screens during the night were excluded from the analysis the association with >2 hours of evening screen use, but not moderate screen use

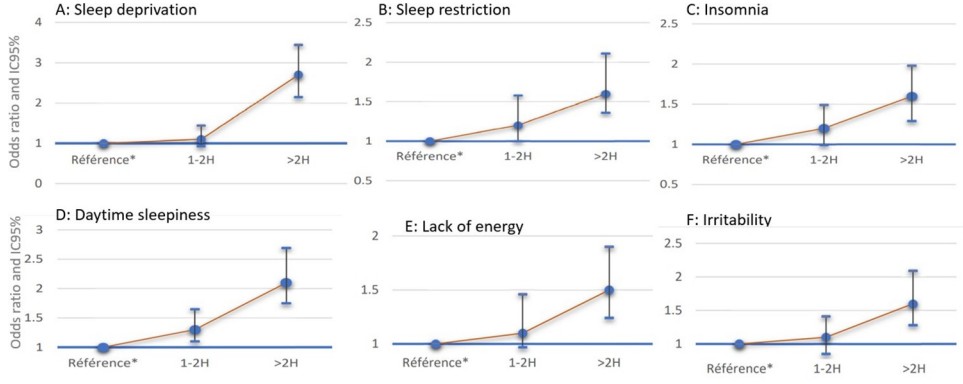

**Fig 1. Duration of evening screen time use and effects.**

**Table 4. The association between the duration of screen exposure in the evening and during the night on sleep and daytime function adjusted for age, sex and sociodemographic class.**

| | Evening screen exposure compared to students using screens for <1 hour* | | | | Night time screen exposure compared to students not using screens at night** | | | | | | | |
| | 1–2 hours screen use N = 641 | | >2 hours screen use N = 549 | | 30 minutes N = 327 | | 30–60 minutes N = 154 | | 60–120 minutes N = 82 | | >120 minutes N = 86 | |
| | Crude OR [IC95%] | Adjusted OR [IC95%] | Crude OR [IC95%] | Adjusted OR [IC95%] | Crude OR [IC95%] | Adjusted OR [IC95%] | Crude OR [IC95%] | Adjusted OR [IC95%] | Crude OR [IC95%] | Adjusted OR [IC95%] | Crude OR [IC95%] | Adjusted OR [IC95%] |
|---|---|---|---|---|---|---|---|---|---|---|---|---|
| Sleep deprivation on schooldays | 1,27 [1,05–1,54] | 1,16 [0,93–1,44] | 2,47 [2,01–3,03] | 2,72 [2,15–3,44] | 2,51 [1,97–3,19] | 2,72 [2,6–3,59] | 2,26 [1,62–3,15] | 2,90 [1,77–3,82] | 3,55 [2,21–5,68] | 5,98 [3,52–10,15] | 3,81 [2,39–6,6] | 5,23 [3,3–9,00] |
| Sleep restriction | 1,32 [1,07–1,62] | 1,27 [1,0–1,58] | 1,73 [1,39–2,14] | 1,69 [1,36–2,11] | 1,76 [1,37–2,27] | 1,72 [1,31–2,25] | 1,82 [1,27–2,61] | 1,92 [1,31–2,80] | 2,54 [1,54–4,18] | 3,17 [1,88–5,35] | 1,91 [1,19–3,4] | 2,05 [1,23–3,42] |
| Insomnia | 0,88 [0,68–1,47] | 0,84 [0,63–1,11] | 1,57 [1,23–2,00] | 1,60 [1,23–2,00] | 2,36 [1,78–3,13] | 2,38 [1,77–3,21] | 2,78 [1,91–3,13] | 2,82 [1,91–4,16] | 4,31 [2,71–6,85] | 4,23 [2,61–6,83] | 5,85 [3,75–9,14] | 5,48 [3,39–8,84] |
| Lack of energy | 1,25 [1,03–1,52] | 1,19 [0,97–1,46] | 1,46 [1,03–1,52] | 1,54 [1,24–1,90] | 1,72 [1,36–2,18] | 1,60 [1,24–2,5] | 2,11 [1,52–2,95] | 2,12 [1,50–3,00] | 3,41 [2,13–5,46] | 3,89 [2,38–6,35] | 2,44 [1,57–3,79] | 2,84 [1,75–4,60] |
| Irritability | 1,08 [0,85–1,37] | 1,10 [0,85–1,41] | 1,52 [1,20–1,93] | 1,64 [1,28–2,09] | 1,71 [1,30–2,26] | 1,75 [1,31–2,33] | 2,99 [2,11–4,24] | 2,76 [1,92–3,96] | 1,70 [1,2–2,84] | 1,66 [0,98–2,82] | 3,92 [2,52–6,11] | 3,56 [2,21–5,75] |
| Daytime sleepiness | 1,41 [1,16–1,71] | 1,35 [1,10–1,65] | 2,15 [1,76–2,64] | 2,17 [1,75–2,69] | 1,80 [1,42–2,28] | 1,67 [1,30–2,15] | 2,43 [1,74–3,39] | 2,59 [1,82–3,68] | 3,16 [1,99–5,1] | 3,81 [2,34–6,21] | 2,92 [1,86–4,56] | 3,05 [1,87–4,95] |
| Poor quality sleep | 1,24 [1,02–1,50] | 1,22 [0,99–1,49] | 1,54 [1,26–1,88] | 1,60 [1,29–1,98] | 1,92 [1,52–2,44] | 1,73 [1,34–2,22] | 2,28 [1,63–3,20] | 2,30 [1,62–3,28] | 2,53 [1,60–4,00] | 2,98 [1,84–4,82] | 1,84 [1,19–2,85] | 2,99 [0,94–3,45] |

Significant data (p < 0.0001) are shown in bold

* Adolescent who use screens for more than 1 hour are compared with those who have an evening screen exposure of less than 1 hour considered as the reference class (N = 1323, OR = 1)

** Adolescent who have nighttime screen exposure are compared with those who never use screen during the night considered as the reference class (N = 1841, OR = 1)

OR ratio are adjusted for age, gender and geographic area of the school as a proxy of socio economic level

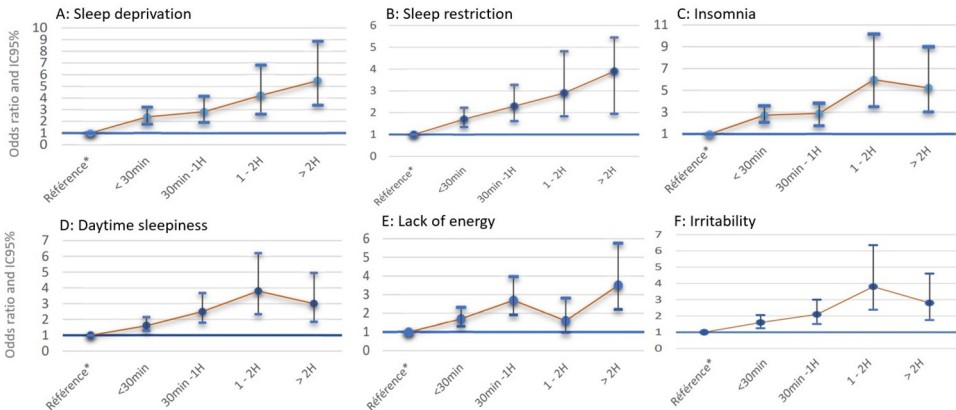

**Fig 2. Duration of night time screen time use and effects.**

was confirmed for sleep restriction OR 1.54 [0.78–1.70], poor quality sleep OR 1.61 [1.23–2.12], insomnia OR 1.54 [1.06–2.23], lack of energy OR 1.21 [0.92–1.60] irritability OR 1.71 [1.23–2.40] and increased daytime sleepiness OR 1.94 [1.47–2.57].

Night-time screen use (Fig 2 and Table 4) showed a clear dose effect across all domains in the fully adjusted model. More than 2 hours of night-time screen use was associated with insomnia OR 5.48 [3.39–8.84], daytime sleepiness OR 3.05 [1.87–4.95], irritability OR 3.56 [2.21–5.75], and lack of energy OR 2.84 [1.75–4.60]. The association with non-refreshing sleep was borderline significant (adjusted OR 2.99 [0.94–3.45]). To examine the possibility that the association with daytime sleepiness was limited to participants with long sleep time (>9 hours per day) the analysis was repeated having excluded long sleepers and showed that the association was reinforced with significant adjusted ORs of 2,00 [1.16–3.46] for 30 minutes use, 3.13 [1.56–6.27] for 30–60 minutes and 6.25 [2.06–18.9] for >120 minutes. The OR in the 60–120 minutes group was not significant but we note low numbers in this group. The analysis was further adjusted for the possession of a screen and for the presence of a screen in the bedroom (see Table 5).

## Discussion

Our study shows a clear association between subjective estimates of evening and night-time screen use, reduced sleep duration, poor sleep and elements affecting daytime functioning (daytime sleepiness, a lack of energy and irritability) in an adolescent population recruited in an educational setting. These effects persisted, especially in the case of night-time screen use, after adjusting for age, sex, school class and sociodemographic status and demonstrated a clear dose effect relationship.

Our findings confirm the results of earlier studies. Insufficient sleep is common in adolescents [21] and multiple studies have confirmed an association not only with screen use but also with the content accessed [26]. A metanalysis [27] of 17 studies in children and adolescents showed a clear association between screen use and inadequate sleep time. A dose response effect between self reported screen use at bedtime and sleep duration was found by Hysing [24] but although screen use during the night is known to be frequent in adolescents in France [28] this is the first study to distinguish the effects of both duration and timing of screen use and to demonstrate a clear dose response relationship.

Nearly all adolescents have access to a screen, and having a screen (not including a smartphone) in the bedroom was frequent: 65% in our study compared to 97% in the USA (NSF

**Table 5. The association between the duration of screen exposure during the evening and at night on sleep and daytime function adjusted for age, sex and sociodemographic class adjusted for screen possession and presence of a screen in the bedroom (adjusted OR only).**

| | Evening screen exposure compared to students using screens for <1 hour* | | Night time screen exposure compared to students not using screens at night ** | | | |
|---|---|---|---|---|---|---|
| | 1–2 hours screen use N = 641 | >2 hours screen use N = 549 | 30 minutes N = 327 | 30–60 minutes N = 154 | 60–120 minutes N = 82 | >120 minutes N = 86 |
| | Adjusted OR [IC95%] | Adjusted OR [IC95%] | Adjusted OR [IC95%] | Adjusted OR [IC95%] | Adjusted OR [IC95%] | Adjusted OR [IC95%] |
| Sleep deprivation on schooldays | 0.87(0.649 1.17) | **2.62 (2.01– 3.42)** | **2.21 (1.61– 3.03)** | **3.07 (2.02– 4.66)** | **5.93 (3.44– 10.21)** | **7.79 (4.48– 13.52)** |
| Sleep restriction | **1,22 (0.99– 1.52)** | **1.60 (1.27– 2.02)** | **1.61 (1.23– 2.11)** | **1.80 (1.23– 2.64)** | **2.90 (1.72– 4.91)** | **1.93 (1.15– 3.24)** |
| Insomnia | 0.84 (0.63– 1.12) | **1.63 (1.25– 2.12)** | **2.42 (1.78– 3.29)** | **2.92 (1.97– 4.32)** | **4.48 (2.75– 7.29)** | **5.62 (3.44– 9.17)** |
| Lack of energy | 1.62 (0.94– 1.43) | **1.48 (1.19– 1.84)** | **1.54 (1.19– 1.99)** | **2.10 (1.49– 2.98)** | **3.98 (2.42– 6.56)** | **2.70 (1.66– 4.40)** |
| Irritability | 1.11 (0.86– 1.43) | **1.65 (1.28– 2.12)** | **1.69 (1.27– 2.27)** | **2.69 (1.87– 3.88)** | 1.65 (0.97– 2.80) | **3.56 (2.19– 5.80)** |
| Daytime sleepiness | **1.32 (1.07– 1.62)** | **2.11 (1.69– 2.63)** | **1.63 (1.27– 2.11)** | **2.49 (1.75– 3.54)** | **3.79 (2.31– 6.22)** | **3.05 (1.86– 5.00)** |
| Poor quality sleep | **1.25 (1.01– 1.53)** | **1.61 (1.29– 2.00)** | **1.73 (1.34– 2.23)** | **2.36 (1.65– 3.37)** | **3.02 (1.86– 4.92)** | **2.14 (1.32– 3.48)** |

Significant data (p < 0.0001) are shown in bold

* Adolescent who use screens for more than 1 hour are compared with those who have an evening screen exposure of less than 1 hour considered as the reference class (N = 1323, OR = 1)

** Adolescent who have a nighttime screen exposure are compared with those who never use screen during the night considered a the reference class (N = 1841, OR = 1)

OR ratio are adjusted for age, gender and geographic area of the school as a proxy of socio economic level screen possession and presence of a screen in the bedroom

sleep in America 2014) and this is increasing over time [29]. Our adjusted model showed that simply having access to a computer or television screen in the bedroom did not increase the risk of negative effects on sleep: it is the duration of screen use and notably the timing of use which determines the effect. Our data enabled us to distinguish evening screen use (after the evening meal but before lights out) and night-time screen use (after lights out). Evening screen use (computer, television or smartphone) was found in 86%, slightly lower than Hysing et als study in 2015 in a population of 16–18 year olds [24], but we note that our population was younger with a mean age of 14.3 years. 27% of our participants regularly used screens during the night and 33% were woken at least once a night by an alert on their mobile phones.

A reduction in sleep time linked to the intensity of screen use during the night reflects the fact that time spent on screens directly reduces sleep time. However, this cannot fully explain the reduction in sleep time observed by those using screens in the evening. Evening screen based activities can be mentally stimulating (for example playing games or interacting on social networks) but screens also expose users to bright, blue enriched light. Evening light exposure has two effects: a directly stimulating effect on the wake systems reducing sleepiness [30] and a potential reduction or abolition of melatonin secretion which shifts the phase of the body clock, leading to later bedtimes and wake times [31]. However this depends on the spectrum of the light exposure (blue light specifically stimulating the melanopsin receptors in the

retina) [10], and also daytime light exposure as bright light exposure during the day can significantly reduce the suppressing effect of evening light on melatonin secretion [32]. It is suggested that the combination of evening light exposure and stimulating screen based activity may explain the later bedtimes noted in evening screen users [26]. All our participants were under 18 and in school and thus obliged to get up early in the school week (mean getting up time was 07:05): early school start times are known to be associated with reduced weekday sleep times in adolescents [33]. A later bedtime in the context of a fixed wake time would reduce total sleep time. At the weekend, high sleep pressure due to reduced sleep on school nights and phase shifted circadian rhythms leading to delayed melatonin offset will lead to later wake times and a difference in sleep duration between the school week and the weekend. This is what we observed, with a clear dose effect relationship between evening screen use and sleep restriction (defined as ≥2 hours more sleep per night on average at the weekend). Sleep deprivation, defined as a sleep time in the week of <7 hours was linked to all night-time screen use and >2 hours of evening screen use.

Screen use has also been shown to impact sleep quality [34]. Increased sleep latency is common in adolescents due to physiological circadian delay [35], so we retained a definition of sleep onset insomnia combining a delay of >60 minutes [24] associated with daytime consequences. We found sleep onset insomnia in 18% of participants and a clear increase in risk with 2 hours evening screen use, confirming the findings of Hysing, Yen and Varghese [24, 36, 37]. Non refreshing sleep affected 45% of our sample and we found a strong association with both >2 hours evening screen use and all night time use. The feeling that sleep is not refreshing has been shown to be linked to inadequate sleep duration [38], but also to sleep fragmentation and difficulties waking in the morning which are linked to circadian phase. Circadian phase shifting with delayed melatonin offset induced by evening/nighttime screen use would lead to being woken up to go to school in the week before melatonin offset. This delayed melatonin offset would cause difficulties waking, which would be exacerbated by the effects of reduced sleep time. We also note that 33% of our population reported being regularly woken at night by telephone alerts which could further fragment sleep.

Evening and night-time screen use also affected daytime functioning. Insufficient sleep is the major cause of excessive daytime sleepiness in adolescents [39, 40]. Daytime sleepiness was measured in our study by the likelihood of falling asleep in class, which represents severe sleepiness, and this was found in 2% of our population. We showed a clear increase in daytime sleepiness linked to both moderate and > 2 hours evening and night-time screen use. More than 2 hours evening use and all levels of night-time screen use were associated with lack of energy during the day and with irritability. Sleepiness has been shown to interfere with learning [40], and irritability reflects difficulties with emotional control in the face of social challenge. Our study was not designed to demonstrate a link between screen use and educational performance, but this has been shown by other studies [22].

Our study was limited by the fact that both sleep habits and screen use depended on self report. However studies using actigraphy have shown broad correlation between subjective and objective reports of sleep timing [41] with higher accuracy in adolescents for sleep onset latency [42]. A more complex problem is that of subjective vs objective screen use. Firstly, as discussed by Kaye et al, there is a clear need for theoretically valid and practically useful conceptualisations of screen time [43]. Estimations of use are affected by the time frame of reference, media multitasking (the number of screens used simultaneously), developmental age [44] and the differentiation between screen time as a numerical measurement (e.g., minutes per day) or "screen use" (number of connections),. Andrew et als study of smartphone use in adults found a reasonable correlation in estimated vs objective use time although a considerable underestimate in the number of brief uses [45] while other studies show only a modest

association between self report and logged screen time and a possible tendency to over report [46]. To our knowledge no study has been able to evaluate objective vs subjective screen time in the context of media multitasking especially in adolescents. Developing objective measures of screen time across multiple devices is necessary but technically challenging. Given the known difficulties in time estimation in the younger population it is likely that adolescents underestimate their screen use.

Our screen use questions were limited to an upper limit of more than 2 hours: given the high number of students (30%) using screens for more than 2 hours in the evening, it would have been useful to have had further detail on high duration screen users: future studies should seek details of upper limit use and we suggest asking about use of 180–240 minutes and >240 minutes. Opticians have started to offer blue light filters on glasses but we did not ask about this, and so cannot identify a subpopulation potentially protected from the adverse effects of blue light on circadian rhythms. The nature of screen based activity may add to the stimulating effect of screens, but our study did not examine content or whether screen use was related to school work or leisure activities, although telephone use is unlikely to be directly related to school work.

Our definition of sleep deprivation as a sleep time <7 hours is less than the 9 hours recommended sleep time for teenagers [47] although this ignores individual variations in sleep needs. It is possible that students who need less that nine hours sleep or who have sleep difficulties use screens when they cannot sleep leading to an association with reduced sleep time. We cannot determine the direction of the effect, but students who need little sleep would not suffer adverse effects on daytime functioning which would potentially reduce the strength of the association. We did not examine potential confounders such as anxiety and depression as the questionnaire was completed online in the classroom setting. Finally our population was not selected randomly as we recruited from schools where teachers had chosen to implement a sleep focussed teaching module. However, within school classes we had an excellent participation rate, ensuring good representation among children and data was adjusted for sociodemographic status estimated by geographical location of the school.

## Conclusion

Our study aimed to explore the role of both duration and timing of screen exposure in adolescents and is first to highlight a dose effect relationship between duration of screen use in a teenage population both in the evening and during the night. Screen use is associated with reduced sleep time, insomnia, non refreshing sleep and daytime consequences such as sleepiness and irritability. Our data suggests that all night-time screen use should be avoided. Moderate evening use of less than 2 hours is not associated with reduced sleep but may still be associated with daytime sleepiness. We suggest that guidelines for the safe use of screens in adolescents should recommend less than 2 hours of screen use in the evening and no screen use at all during the night.

## Supporting information

**S1 Data.**
(XLS)

## Author Contributions

**Conceptualization:** Sarah Hartley, Sylvie Royant-Parola.

**Data curation:** Sarah Hartley, Sylvie Royant-Parola, Bobette Matulonga.

**Formal analysis:** Ayla Zayoud, Bobette Matulonga.

**Investigation:** Sarah Hartley, Sylvie Royant-Parola, Ayla Zayoud, Bobette Matulonga.

**Methodology:** Sarah Hartley, Sylvie Royant-Parola, Bobette Matulonga.

**Project administration:** Sylvie Royant-Parola, Isabelle Gremy.

**Resources:** Sylvie Royant-Parola, Isabelle Gremy.

**Supervision:** Isabelle Gremy, Bobette Matulonga.

**Validation:** Sarah Hartley, Sylvie Royant-Parola, Isabelle Gremy, Bobette Matulonga.

**Visualization:** Sarah Hartley, Sylvie Royant-Parola, Ayla Zayoud.

**Writing – original draft:** Sarah Hartley.

**Writing – review & editing:** Sarah Hartley, Sylvie Royant-Parola, Ayla Zayoud, Isabelle Gremy, Bobette Matulonga.

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
