## [Decision Letter · Decision Letter 0]

30 May 2022

PONE-D-21-39539Do both timing and duration of screen use affect sleep patterns in adolescents?PLOS ONE

Dear Dr. Hartley,

Thank you for submitting your manuscript to PLOS ONE. After careful consideration, we feel that it has merit but does not fully meet PLOS ONE’s publication criteria as it currently stands. Therefore, we invite you to submit a revised version of the manuscript that addresses the points raised during the review process.

I would kindly ask you to revise the manuscript following all of the reviewer's comments. Importantly, the fact that there are no objective measurements makes the research difficult to assess. As a consequence, I would kindly ask you to add "self-reported" in front of "screen time" and discuss specifically the limitations of using self reports in this case. See DOI: 10.3390/ijerph17103661 for a discuss on these topics.

We look forward to receiving your revised manuscript.

Kind regards,

Manuel Spitschan

Academic Editor

PLOS ONE

Journal Requirements:

“The Réseau Morphée and the ORS are funded by the Paris region health authority (ARS Ile de France). The ORS is additionally funded by the Paris region (Conseil Régional Ile de France). No specific grant for the study was obtained”.

“The Réseau Morphée and the ORS are funded by the Paris region health authority (ARS Ile de France). The ORS is additionally funded by the Paris region (Conseil Régional Ile de France).

The authors received no specific funding for this work.”

Additional Editor Comments (if provided):

I apologise for the delay in getting back to you regarding your manuscript. It has been immensely difficult to find reviewers at this time. Many are unresponsive, and a total of 11 reviewers declined to review the manuscript.

Reviewers' comments:

Reviewer's Responses to Questions

**Comments to the Author**

1. Is the manuscript technically sound, and do the data support the conclusions?

Reviewer #1: Partly

2. Has the statistical analysis been performed appropriately and rigorously? 

Reviewer #1: Yes

3. Have the authors made all data underlying the findings in their manuscript fully available?

Reviewer #1: Yes

4. Is the manuscript presented in an intelligible fashion and written in standard English?

Reviewer #1: Yes

5. Review Comments to the Author

Reviewer #1: Review: Do both timing and duration of screen use affect sleep patterns in adolescents

The authors investigate the involvement of screen usage in sleep disturbances in adolescents. A dose-response curve between levels of screen use on sleep deprivation and restriction was shown. The authors conclude that both screen use and timing are associated with adverse effects on sleep and daytime functioning. Overall, the study is interesting and valuable. However, the study design and interpretation does come with some clear limitations, as only subjective measurements and arbitrary cutoff points were used.

Major concerns

- Even though this is addressed in the discussion section of the manuscript, the biggest limitation of the study is that there are no objective measurements. As the authors address, it is difficult to estimate how well aware these adolescents are of their own screen usage. Even though the authors do state that there a modest association between self-report and logged screen time, can they at least speculate on the fact if these results might be over or under estimated?

- The cutoff points for various definitions are arbitrary. For example, excessive screen use is defined as >3 hours a day of screen use for social activities. If you use a different cutoff point, the results/interpretation might be completely different. I think the authors should elaborate on why certain thresholds were chosen, so that the reader for themselves can decide if they agree with these thresholds or not (for example “>3 hours defined as excessive was based upon a histogram distribution of all screen times, indicating that the top 5% was 3 hours or more”).

- The authors state that there is a dose-response relationship, but I have not seen a graph depicting this relationship. The authors should add this.

Abstract:

- “Sleep habits on school nights and weekends, symptoms of insomnia and daytime repercussions were recorded.” – please specify how

Implications et contribution:

- Please change to “Implication and contribution”

Introduction:

- I would suggest to rephrase “Sleep is regulated by the homeostatic and the circadian systems” to “Sleep is regulated by homeostatic sleep pressure and the circadian drive for sleep”, as homeostatic sleep pressure is not a system

- “the latter depends on external synchronizers or zeitgebers” – ref

- “For sleep to be restorative, the homeostatic and circadian systems need to be synchronized, and sleep time needs to be adequate” – this is not true, we actually don’t really know what constitutes restorative sleep. This may significantly differ from what is defined as restorative sleep as well, since objective and subjective measures differ. Please rephrase.

Materials and methods:

- “Daytime sleepiness was defined as a score >6 on a visual analogue scale” – is there a ref for this as well? Or is this just defined by the researchers?

- The materials and methods do not describe at all how the participants were separated into the different groups, while listed in the abstract as “(≥2 hours difference in sleep duration on school nights vs weekends)”. I am wondering what that cutoff is based upon and if that is a reasonable way of dividing people up.

Results

- Population and Sleep habits – please indicate median [IQR range] for demographic variables. That’s much more appropriate than the mean. Moreover, SD’s are not reported in the text.

- Also for table 1, I would recommend to report the median [IQR] since that is more appropriate and informative for population values

- “95.4% got up between 06:00 and 08:00 on school days, whereas 10% got up after 13:00 at the

Weekends” I wonder why the authors choose to name these specific values. To make a good comparison, would you not rather report when 95.4% got up on the weekend?

- “Before bedtime 11% read or listened to music, 30% did passive screen-based activities (ex. watching a film) 32% did active screen based activities (social networks, video games) and 27% did none of the above. A bedtime after midnight was much more common at the weekend (62% vs 14%).” – when you report that at this location, I also would like to know the duration of screen time usage, but in paragraph screen use such things are reported. I would therefore suggest to move this section.

- “Sleep restriction, defined as a reduction of >2 hours in sleep time on schooldays” – I think this should be defined in the materials and methods. Moreover, I don’t know how the authors came up with this cutoff point. Has that been previously reported in literature? Or is this a made-up cut-off point?

- “was intense screen use during the night” – but it has not been defined by the authors what is intense screen use. Also, for here, I wonder if this is based upon literature or not. If not, it should be stated clearly that these cut-off points have been designed by the authors.

- Table 3 & 4: could you add a column with the statistics?

- Since the authors claim there is a dose-response relationship, I would like to actually see a dose-response curve. The article would benefit significantly from a graph depicting the relationship between screen time and sleep and/or daytime functioning

- I’d like to see more results concerning the content. For me, this part is particularly interesting, since it can come with a lot of consequences. Can the authors show more of these data?

Discussion:

- “abolition of melatonin secretion which shifts the phase of the body clock” – well this significantly depends on the light intensity and spectral distribution of the screen. Please phrase more carefully

- “(our mean getting up time was 07:05)” – rephrase, since it was not you who got up at that time but the participants of the study.

- “and phase shifted circadian rhythms leading to delayed melatonin offset” – since you did not measure melatonin, you did not measure this. Therefore, you should rephrase. You’re not even sure that this is happening in this population.

- “The stimulating effects of light, modified melatonin secretion but also the mental stimulation of screen based activities may delay bedtimes” – again, this was not measured so please formulate more carefully

- “consequences. We found sleep onset insomnia in 16% of participants and a clear..” – why was this not described in the results section? You cannot report new results in the discussion. Please describe these results in the result section.

6. PLOS authors have the option to publish the peer review history of their article (what does this mean?). If published, this will include your full peer review and any attached files.

Reviewer #1: No

---

## [Author Response · Author response to Decision Letter 0]

29 Jun 2022

We have responded to the comments point by point in the document below and have submitted a revised version of the article marked up in red and a version without marks. 

 

Review: Do both timing and duration of screen use affect sleep patterns in adolescents

The authors investigate the involvement of screen usage in sleep disturbances in adolescents. A dose-response curve between levels of screen use on sleep deprivation and restriction was shown. The authors conclude that both screen use and timing are associated with adverse effects on sleep and daytime functioning. Overall, the study is interesting and valuable. However, the study design and interpretation does come with some clear limitations, as only subjective measurements and arbitrary cutoff points were used. 

Major concerns 

- Even though this is addressed in the discussion section of the manuscript, the biggest limitation of the study is that there are no objective measurements. As the authors address, it is difficult to estimate how well aware these adolescents are of their own screen usage. Even though the authors do state that there a modest association between self-report and logged screen time, can they at least speculate on the fact if these results might be over or under estimated? 

- 

- We would like to thank the reviewer for this point which we have raised in the discussion but which merits further exploration. We have enlarged the discussion as follows. 

A more complex problem is that of subjective vs objective screen use. Firstly, as discussed by Kaye et al, there is a clear need for theoretically valid and practically useful conceptualisations of screen time[39]. Estimations of use are affected by the time frame of reference, media multitasking (the number of screens used simultaneously), developmental age [40] and the differentiation between screen time as a numerical measurement (e.g., minutes per day) or “screen use” (number of connections),. Andrew et als study of smartphone use in adults found a reasonable correlation in estimated vs objective use time although a considerable underestimate in the number of brief uses [41] while other studies show only a modest association between self report and logged screen time and a possible tendency to over report [42]. To our knowledge no study has been able to evaluate objective vs subjective screen time in the context of media multitasking especially in adolescents. Developing objective measures of screen time across multiple devices is necessary but technically challenging. Given the known difficulties in time estimation in the younger population it is likely that adolescents underestimate their screen use.

- 

- The cutoff points for various definitions are arbitrary. For example, excessive screen use is defined as >3 hours a day of screen use for social activities. If you use a different cutoff point, the results/interpretation might be completely different. I think the authors should elaborate on why certain thresholds were chosen, so that the reader for themselves can decide if they agree with these thresholds or not (for example “>3 hours defined as excessive was based upon a histogram distribution of all screen times, indicating that the top 5% was 3 hours or more”). 

We agree with the reviewer. Our initial questionnaire used the following estimations of screen use (No use (>30 minutes, 30-60 minutes, 1-2 hours, > 2 hours) which we condensed for evening use but kept for nighttime use. As 30% of our population used screens for more than 2 hours in the evening, we should have included a higher estimate. This is discussed in the limitations section

Our screen use questions were limited to an upper limit of more than 2 hours: given the high number of students (30%) using screens for more than 2 hours in the evening, it would have been useful to have had further detail on high duration screen users: future studies should seek details of upper limit use and we suggest asking about use of 180-240 minutes and >240 minutes.

The histograms of use are shown below

- The authors state that there is a dose-response relationship, but I have not seen a graph depicting this relationship. The authors should add this. 

We propose two figures which show the increase in the OR for different outcome measures for increasing exposure to screens for evening and nighttime use and which clearly show the dose response relationship for increasing screen time

Figure 1: Duration of screen use and effects of sleep, daytime sleepiness, energy and irritability in the evening (1A-F) 

Figure 2: Duration of screen use and effects of sleep, daytime sleepiness, energy and irritability in the night (1A-F) 

Abstract: 

- “Sleep habits on school nights and weekends, symptoms of insomnia and daytime repercussions were recorded.” – please specify how 

- The text has been corrected 

- Cross sectional questionnaire survey of adolescents recruited in schools. Sleep habits on school nights and weekends, symptoms of insomnia and daytime repercussions were recorded using an online questionnaire administered in the classroom setting.

Implications et contribution: 

- Please change to “Implication and contribution”

- The text has been corrected 

- Implication et contribution

Introduction: 

- I would suggest to rephrase “Sleep is regulated by the homeostatic and the circadian systems” to “Sleep is regulated by homeostatic sleep pressure and the circadian drive for sleep”, as homeostatic sleep pressure is not a system 

- The text has been corrected

- Sleep is regulated by homeostatic sleep pressure and the circadian drive for sleep

- “the latter depends on external synchronizers or zeitgebers” – ref

- Moore’s article from 1997 has been referenced

- 

- “For sleep to be restorative, the homeostatic and circadian systems need to be synchronized, and sleep time needs to be adequate” – this is not true, we actually don’t really know what constitutes restorative sleep. This may significantly differ from what is defined as restorative sleep as well, since objective and subjective measures differ. Please rephrase. 

- 

- We thank the reviewer for highlighting this imprecise summary: we have rephrased as follows. 

- Both sleep duration and the phase of the circadian system are important for sleep timing, sleep quality and daytime vigilance (Daan). 

Materials and methods: 

- “Daytime sleepiness was defined as a score >6 on a visual analogue scale” – is there a ref for this as well? Or is this just defined by the researchers? 

- We did not use a validated scale. The midpoint of our scale was 5 (neither sleepy nor particularly vigilant) with a minimum of 0 (very awake in class) and a maximum of 10 (extremely sleepy in class): we considered that a score above the midpoint represented daytime sleepiness in class.

- We have modified the text as follows:

- Daytime sleepiness was defined as a score >6 on a visual analogue scale of 1-10 concerning the likelihood of falling asleep in class with a maximum score of 10 (extremely sleepy in class).

- 

- The materials and methods do not describe at all how the participants were separated into the different groups, while listed in the abstract as “(≥2 hours difference in sleep duration on school nights vs weekends)”. I am wondering what that cutoff is based upon and if that is a reasonable way of dividing people up. 

- We divided our participants into groups based on their screen exposure. This is explained in the statistical analysis section

- For multivariant analysis, evening screen use with 5 possible responses from 0 minutes to >120 minutes were collapsed into 3 groups (<60 minutes, 60-120 minutes and >120 minutes). For night-time screen use, the original 5 groups (0 minutes to >120 minutes) were retained as we considered it possible that lower duration screen exposure during the night was physiologically important.

- The reviewer is referring to our definition of sleep restriction which is a difference of >2 hours between sleep time in the week and sleep time at the weekend and was examined as one of the potential impacts on sleep and sleep timing of screen use. We defined sleep restriction as >2 hours per night following Lo et al’s study in 2019 which found an impact of sleep restriction of 2,5 hours less than ideal sleep time (defined by Lo as 9 hours). We compared sleep deprivation (sleep time of <7 hours) with sleep restriction (a difference between sleep time at the weekend and during the week) to attempt to compensate for adolescents whose sleep needs were less than 9 hours. This has been clarified in the text

- We compared sleep deprivation (sleep time of <7 hours) with sleep restriction (a difference between sleep time at the weekend and during the week of>2 hours) to attempt to compensate for adolescents whose sleep needs were less than 9 hours following the definition of sleep restriction in the study by Lo et al [23]. 

Results

- Population and Sleep habits – please indicate median [IQR range] for demographic variables. That’s much more appropriate than the mean. Moreover, SD’s are not reported in the text. 

- We thank the reviewer for this helpful comment on the presentation of our results: we have revised the descriptive statistics and analysis used and modified the text accordingly both in the statistical analysis section:

Quantitative variables (sleep duration, bedtime and getting up time) were described by mean median and standard deviation interquartile range (IQR) and were compared using the Kruskal-Wallis test.

And in the results section: note that SD are reported in hours (so 1:52 = 1 hour and 52 minutes)

- 2513 students were included: 53.4% were girls with a mean age of 14.3 years (median :15 years, IQR: [12, 16]). 

- Mean sleep time over the entire population was 7.8 hours (median 8 hours; IQR: [7,9]; SD 1:52) during the school week and 9.75 hours (median 10 hours; IQR: [9,11]; SD 2:17) at the weekend:

- Also for table 1, I would recommend to report the median [IQR] since that is more appropriate and informative for population values 

- We thank the reviewer for this helpful comment on the presentation of our results: we have modified the table accordingly: this is the full table with the deletions in strikethrough, the modified table is shown without strike through in the text

 <12 years

n=847 12-15 years

n=842 >15 years

n=816 Total

n=2513 p P tendance

Mean sleep time on school days 08:46(1:51) 07:36(1:42) 07:03(1:36) 7:49 <0.0001 <0.0001

Median Sleep time on school days; hours: minutes (SD) 9:00(1:51) 8:00 (1:42) 7:00 (1:36) 8:00 (1:52) <0.0001 <0.0001

Mean sleep time on weekends 10 :02(2 :34) 09 :46(1 :58) 09 :25 (2 :10) 9:45 <0.0001 <0.0001

Median sleep time on weekends ; 

hours: minutes (SD) 10:05 (2:35) 10:04 (1:58) 9: 57(2:11) 10:00 (2:17) <0.0001 <0.0001

Sleep deprivation on schooldays 9% 21% 31% 20% <.0001 <.0001

Sleep restriction 30% 45% 46% 41% <.0001 <.0001

Insomnia 19% 17% 18% 18% 0.45 0.87

Difficulty falling asleep 20% 14% 16% 16% 0.004 0.02

Daytime sleepiness 3% 2% 1% 2% 0.043 0.012

Poor quality sleep 36% 48% 52% 45% <.0001 <.0001

- 

- “95.4% got up between 06:00 and 08:00 on school days, whereas 10% got up after 13:00 at the

Weekends” I wonder why the authors choose to name these specific values. To make a good comparison, would you not rather report when 95.4% got up on the weekend? 

- 

- We agree that this is poorly formulated, and have enlarged the explanation. The actual figures are as follows (in percentages) for the weekend getting up times

- 1.75% 6-7 am

- 4.% 7-8 am

- 9.35% 8-9 am

- 20.25% 9-10am

- 24.91% 10-11am

- 18.86% 11-12am

- 11.30% 12-1pm

- 5.2 % 1-2pm

- 3.82% 2 pm

We have modified the text as follows

95% got up between 06:00 and 08:00 on school days, whereas 94% got up after 08:00 at the weekends (with 36,3% after 11H00 and 10% after 13:00).

- “Before bedtime 11% read or listened to music, 30% did passive screen-based activities (ex. watching a film) 32% did active screen based activities (social networks, video games) and 27% did none of the above. A bedtime after midnight was much more common at the weekend (62% vs 14%).” – when you report that at this location, I also would like to know the duration of screen time usage, but in paragraph screen use such things are reported. I would therefore suggest to move this section. 

- We agree with the reviewer and the section has been moved to the screen use chapter

- “Sleep restriction, defined as a reduction of >2 hours in sleep time on schooldays” – I think this should be defined in the materials and methods. Moreover, I don’t know how the authors came up with this cutoff point. Has that been previously reported in literature? Or is this a made-up cut-off point? 

- We have added this into the methods section, please see above

- “was intense screen use during the night” – but it has not been defined by the authors what is intense screen use. Also, for here, I wonder if this is based upon literature or not. If not, it should be stated clearly that these cut-off points have been designed by the authors. 

- We agree with the reviewer: the phrase ‘intense screen use is potentially misleading: it represents the extreme of our original 5 groups from the questionnaire ( 0 minutes, <30 minutes, <1 hour, 1-2 hours, <2 hours). We have modified the text at every point where it appears to ‘>2 hours’ or ‘more than 2 hours’

- 51% used screens for >120 minutes in the evening

- 

- Table 3 & 4: could you add a column with the statistics? We have reviewed the tables. Adding separate columns with the statistics led to a table that was unreadable as there were 24 columns. Following the presenation in other similar papers we show significant p values (ie p<0.0001) in bold and we have clarified this in the legend and in the text. 

- In the text

- (all data in bold are significant with p < 0.0001).

- In legend of figure 4 and 5: 

- Significant data (p < 0.0001) are shown in bold

- Since the authors claim there is a dose-response relationship, I would like to actually see a dose-response curve. The article would benefit significantly from a graph depicting the relationship between screen time and sleep and/or daytime functioning 

- This has been added (see above)

- I’d like to see more results concerning the content. For me, this part is particularly interesting, since it can come with a lot of consequences. Can the authors show more of these data? 

- We have very little data concerning content and concerns about data quality. In practice multiple contents are used during a single period of screen use, often simultaneously. Our questionnaire only asked about the principal use and did not provide the possibility of recording several activities. We focused on screen time, not screen based activities. 

- 

- Discussion: 

- “abolition of melatonin secretion which shifts the phase of the body clock” – well this significantly depends on the light intensity and spectral distribution of the screen. Please phrase more carefully 

- We thank the reviewer for this pertinent comment: the text has been rephrased as below

- Evening light exposure has two effects: a directly stimulating effect on the wake systems reducing sleepiness [31] and a reduction or abolition of melatonin secretion which shifts the phase of the body clock, leading to later bedtimes and wake times [32], although this depends on the spectrum of the light exposure (blue light specifically stimulating the melanopsin receptors in the retina)[10] and also daytime light exposure[33]

- “(our mean getting up time was 07:05)” – rephrase, since it was not you who got up at that time but the participants of the study. 

- The text has been revised

- (mean getting up time was 07:05)

- “and phase shifted circadian rhythms leading to delayed melatonin offset” – since you did not measure melatonin, you did not measure this. Therefore, you should rephrase. You’re not even sure that this is happening in this population. 

- 

- The reviewer is absolutely right, our phrasing of a hypothesis to explain reduced sleep times (later bedtime and a fixed getting up time) was sloppy. The text has been rewritten

- It is suggested that the combination of evening light exposure and stimulating screen based activity may explain the later bedtimes noted in evening screen users[34]. All our participants were under 18 and in school and thus obliged to get up early in the school week (mean getting up time was 07:05): early school start times are known to be associated with reduced weekday sleep times in adolescents [35]. A later bedtime in the context of a fixed wake time would reduce total sleep time.

- 

- “The stimulating effects of light, modified melatonin secretion but also the mental stimulation of screen based activities may delay bedtimes” – again, this was not measured so please formulate more carefully 

- We have reformulated as follows to remove the reference to potential mecanisms

- Screen use has also been shown to impact sleep quality[35]

- “consequences. We found sleep onset insomnia in 16% of participants and a clear..” – why was this not described in the results section? You cannot report new results in the discussion. Please describe these results in the result section. 

The details about the prevalence of insomnia are reported in table 1 but we note an error in the text: the correct figure is 18%: this has been corrected. 

We found sleep onset insomnia in 18% of participants and a clear increase in risk with > 2 hours evening screen use, confirming the findings of Hysing, Yen and Varghese [24,37,38].

---

## [Decision Letter · Decision Letter 1]

4 Oct 2022

Do both timing and duration of screen use affect sleep patterns in adolescents?

PONE-D-21-39539R1

Dear Dr. Hartley,

We’re pleased to inform you that your manuscript has been judged scientifically suitable for publication and will be formally accepted for publication once it meets all outstanding technical requirements.

Kind regards,

Manuel Spitschan

Academic Editor

PLOS ONE

Additional Editor Comments (optional):

Reviewers' comments:

Reviewer's Responses to Questions

**Comments to the Author**

1. If the authors have adequately addressed your comments raised in a previous round of review and you feel that this manuscript is now acceptable for publication, you may indicate that here to bypass the “Comments to the Author” section, enter your conflict of interest statement in the “Confidential to Editor” section, and submit your "Accept" recommendation.

Reviewer #1: All comments have been addressed

2. Is the manuscript technically sound, and do the data support the conclusions?

Reviewer #1: Yes

3. Has the statistical analysis been performed appropriately and rigorously? 

Reviewer #1: Yes

4. Have the authors made all data underlying the findings in their manuscript fully available?

Reviewer #1: Yes

5. Is the manuscript presented in an intelligible fashion and written in standard English?

Reviewer #1: Yes

6. Review Comments to the Author

Reviewer #1: The authors have addressed all my comments and concerns, and with these changes, I think the manuscript should be acceptable for publication.

7. PLOS authors have the option to publish the peer review history of their article (what does this mean?). If published, this will include your full peer review and any attached files.

Reviewer #1: No

---

## [Editor Report · Acceptance letter]

12 Oct 2022

PONE-D-21-39539R1 

Do both timing and duration of screen use affect sleep patterns in adolescents? 

Dear Dr. Hartley:

I'm pleased to inform you that your manuscript has been deemed suitable for publication in PLOS ONE. Congratulations! Your manuscript is now with our production department. 

Kind regards, 

on behalf of

Dr. Manuel Spitschan 

Academic Editor

PLOS ONE